# Microscopical Evaluation of Smears of the Leptomeninges to Predict Meningitis in Piglets

**DOI:** 10.3390/vetsci9070341

**Published:** 2022-07-05

**Authors:** Marc Schyns, Dominiek Maes, Wikke Kuller, Erik Weerts

**Affiliations:** 1Business Unit Intensive Livestock, MSD Animal Health Benelux, Wim de Körverstraat 35, 5830 AA Boxmeer, The Netherlands; 2Unit of Porcine Health Management, Faculty of Veterinary Medicine, Ghent University, Salisburylaan 133, 9820 Merelbeke, Belgium; dominiek.maes@ugent.be; 3University Farm Animal Practice, Reijerscopse Overgang 1, 3481 LZ Harmelen, The Netherlands; wkuller@ulp.nu; 4Division of Pathology, Department Biomolecular Health Sciences, Faculty of Veterinary Medicine, Utrecht University, Yalelaan 1, 3584 CL Utrecht, The Netherlands; e.a.w.s.weerts@uu.nl

**Keywords:** *Streptococcus suis*, meningitis, pathological examination, cytological evaluation

## Abstract

**Simple Summary:**

Meningitis (inflammation of the membranes covering the brain) is a common problem in 3- to 10-week-old piglets. It is often caused by bacteria that are called Streptococcus. Clinical signs such as problems with coordination of the limbs can be observed by the farmer, but it is also possible that piglets die suddenly, before clinical signs are detected. Therefore, examining dead piglets is often performed to establish a diagnosis. Unfortunately, meningitis is often difficult to recognise by the eye. To assist veterinarians in forming a quick diagnosis of meningitis and possible bacterial involvement, a microscopic evaluation of cells taken from the meninges was performed. This examination allowed the prediction of the likely presence of meningitis and the involvement of Streptococcus species in most of the cases. Microscopic findings were compared with bacteriological culture results, and based on this comparison, indications of meningitis and Streptococcal presence seemed correct in 89% and 100% of the cases, respectively. However, if only a few cells indicating meningitis were observed microscopically, no reliable prediction was possible. In conclusion, microscopic evaluation of cells from the meninges of piglets can help swine practitioners in establishing a tentative diagnosis of meningitis more quickly, which allows for better treatment, improves animal welfare, and reduces antimicrobial use.

**Abstract:**

Meningitis, caused by bacterial infections such as *Streptococcus* spp., is a frequently observed cause of death in pigs. In order to implement effective treatment and avoid further losses, it is important to establish this diagnosis quickly. However, this is often difficult because macroscopic lesions may not be visible, and additional laboratory testing may take several days. The present study investigated whether microscopical examination of impression smears of the leptomeninges taken during necropsy can help in establishing a presumptive diagnosis of meningitis in pigs more quickly. In total, 54 pigs suffering from neurological signs and/or (acute) mortality were examined. They were 3 to 10 weeks old and originated from 16 farms. From each pig, bacterial culture was performed on one half of the brain using a swab from the leptomeninges. From the other half, paired cytological impression smears of the leptomeninges were stained with a commercial quick stain dye (Hemacolor^®^) and Gram stain and microscopically evaluated for the abundance of neutrophils and the presence of short-chain coccoid bacteria. Bacterial culture of the leptomeninges was positive in 36/54 cases, in 28 of which *Streptococcus* spp. were found. The numbers of smears with low, moderate, or high abundance of neutrophils were 19, 17, and 18, respectively. Short-chain coccoid bacteria were detected successfully in 12 pigs in the Gram-stained smear. The positive predictive value of smears with moderate or high abundance of neutrophils for bacterial presence and, therefore, likely meningitis was 89%, whereas the negative predictive value of smears with low abundance of neutrophils was 74%. The positive predictive value of smears with short chains of coccoid bacteria for diagnosis of *Streptococcus* spp. was 100%, whereas the negative predictive value was 62%. In conclusion, microscopical examination of impression smears of the leptomeninges of piglets with neurological signs and/or (acute) mortality is a feasible procedure that can help swine practitioners in establishing a tentative diagnosis of meningitis more quickly, especially if neutrophils are abundant, and short chains of coccoid bacteria are present.

## 1. Introduction

Neurological signs in suckling and nursery pigs are common in pig farms all over the world. In many cases, they are caused by purulent meningitis, one of the most important and also life-threatening disease conditions in piglets. On other occasions, neurological signs may have different pathogeneses, e.g., in cases of oedema disease or deprivation of water. Purulent meningitis is mostly caused by bacterial infections such as *Streptococcus* spp., *Glaesserella parasuis* (*G. parasuis*), and *E. coli.* Infections with *Streptococcus* spp. are considered the most frequent cause of purulent meningitis, and from this bacterial family, *Streptococcus (S.) suis* is the most important pathogen. This bacterium in the acute phase causes a variety of mostly purulent inflammatory lesions such as meningitis, polyarthritis, and endocarditis, and these infections can lead to significant mortality [1]. In the Netherlands, *S. suis* is the foremost cause of clinical problems in suckling and nursery piglets [2]. As a result, a major part of the antimicrobial usage in the Dutch pig sector is related to *S. suis* infections [3], especially to control meningitis in newly weaned pigs. Other Streptococcal species such as *S. dysgalactiae* subsp. *equisimilis* and *S. porcinus* can also incidentally lead to problems [1]. *S. dysgalactiae* subsp. *equisimilis* may induce lesions similar to those caused by the various *S. suis* subtypes, whereas *S. porcinus* is mostly associated with so-called ‘jowl abscesses’ or as a cause of septicemia.

Establishing a diagnosis of (per)acute purulent meningitis purely based on clinical signs or by means of macroscopic tissue examination during necropsy is usually difficult or unreliable. Therefore, additional laboratory testing is needed. Bacterial culture is often used to come to a presumptive diagnosis of bacterium-induced meningitis via identification of the etiologic agent. However, bacterial culture takes time, as the outcome is often known only 2–5 days later. During this timeframe, in an outbreak, many more new cases of sepsis and consequential meningeal infection can occur, leading to significant animal suffering, mortality, and economic losses. Therefore, an additional screening tool to assess and predict the presence of meningitis and ideally also the presence of bacteria without the need to wait for culture results or to send to an external laboratory may be very helpful during the diagnostic process. Such a tool could also be beneficial to establish a more effective first treatment during an earlier stage of the disease process. This may ultimately lead to a lower overall antimicrobial usage and less animal suffering.

Under normal circumstances, leptomeninges contain only a minimal number of inflammatory cells, and especially neutrophilic granulocytes are rarely present [4]. Bacterial infection of the meninges in the (per)acute phase usually induces purulent inflammation, which is microscopically characterised by infiltration of neutrophilic granulocytes [1,5]. The presence of neutrophilic granulocytes in a cytological impression smear of the intact leptomeningeal surface will, therefore, be indicative of a (per)acute purulent meningitis, and in these cases, the involvement of bacteria, e.g., *Streptococcus* spp. as potential aetiology is to be expected with a much higher probability. In addition, cytologic smears may also offer the possibility to directly visualise the bacterial agent. In human medicine, microscopic Gram stain examination of cerebrospinal fluid is often used to determine the presence of a bacterial agent in cases in which meningitis is suspected [6]. This method has been proven useful for the quick diagnosis of bacterial infection, and it has been reported to be a relatively sensitive and specific method [6]. Microscopic Gram stain examination could, therefore, potentially also be of use in smears from swine leptomeninges to predict meningitis and to differentiate among several bacteria based on morphology and staining characteristics [7].

The present study investigated whether microscopic examination of cytological impression smears of the leptomeningeal surface, taken during necropsy of piglets with neurological signs and/or (acute) mortality, is a useful tool in establishing a quick, presumptive diagnosis of purulent meningitis. These smears of the leptomeninges were evaluated for the abundance of neutrophilic granulocytes and the presence of short chains of coccoid bacteria.

## 2. Materials and Methods

### 2.1. Study Animals

In total, 54 piglets from diagnostic cases that originated from 16 client farms of the veterinary practise Someren (Someren, The Netherlands) were included in this study. The pigs were sent for necropsy in the veterinary practise from November 2016 to April 2019. The number of pigs per farm was as follows: one pig per farm (*n* = 2), two (*n* = 5), three (*n* = 4), four (*n* = 3), six (*n* = 1), and twelve (*n* = 1). The age of the pigs ranged from 3 to 10 weeks of age. In total, 3 suckling piglets and 51 weaned piglets were selected. The main reasons for sending piglets to the clinic were the presence of neurological signs and higher mortality or sudden death. The pigs either had died suddenly or were euthanised on the farm (the results of individual animals are provided in Appendix A). Antibiotic treatment in the week before death or euthanasia was an exclusion criterion. The piglets were transported under cooled conditions (between 2 and 8 °C) by the herd veterinarian or by a courier. Necropsy was performed by the veterinarian on the same day that the piglets arrived at the clinic.

### 2.2. Necropsy and Sampling

A necropsy was performed on each pig, and macroscopical findings were recorded. The skull was split sagittally by sawing. Then, the brain was removed from one of the halves of the skull with a clean scissor. This was carried out carefully to prevent contamination of the leptomeninges. From one-half of the brain, bacterial culture was performed based on a swab from the leptomeninges. From the other half, two impression smears for cytological examination were taken by gently pressing microscopic slides on random spots of the intact leptomeningeal surface. Apart from infrequent, minimal-to-mild prominence of the meningeal vasculature due to hyperaemia, none of the brains macroscopically presented with clear lesions. As it is often conventional during necropsies performed by swine practitioners under normal diagnostic practise, no brain samples were fixed in formalin for additional histologic examination.

### 2.3. Bacterial Culture

The sample of the leptomeninges was cultured in blood agar and MacConkey agar, and plates were visually inspected every day for at least 72 h after the start of the culture. The bacterial culture was visually inspected, and no distinction was made between different species of *Streptococcus* spp. In cases in which polyserositis due to *G. parasuis* was suspected, an additional culture of the leptomeninges in a medium of chocolate blood agar and under low oxygen conditions was performed.

### 2.4. Cytological Examination of Smears of the Leptomeninges

The first cytology smear was stained with the Hemacolor^®^ Rapid staining of the blood smear (In Vitro Diagnostic Medical Device) and evaluated microscopically for the presence of neutrophils, i.e., cells with a polymorphic, multilobular nucleus (>2 lobes) and without visible cytoplasmic granules. They were classified into the following three groups, based on the abundance of neutrophils (Figure 1):

Low: maximum of 5 neutrophils per 400× magnification field, only found after thorough searching. Other visible cells only comprised monomorphous, flat spindeloid to sometimes slightly cuboidal cells which were regularly surrounded by some faint fibrillar extracellular matrix (fibroblasts with collagen) (Figure 1a,b). No other clusters of potential mononuclear inflammatory cells were visible.Moderate: amid areas with fibroblasts and collagen, small focal to multifocal clusters of neutrophils (5–100 per 400× magnification field), mostly only spotted after some searching through the slide. Sometimes, clusters of heterogeneous round cells were observed (Figure 1c,d) that did not resemble the monomorphous flat to cuboidal fibroblastic cells (likely mononuclear inflammatory cells).High: large numbers (>100, ‘uncountable’) of neutrophils already visible with low magnification (40–100×) in almost every frame viewed (Figure 1e,f). The presence of other cell types (mononuclear inflammatory cells, fibroblasts) was generally obscured due to the large numbers of neutrophils.

The second smear was stained with the Gram staining (Pro-Lab Diagnostics) and evaluated via light microscopy for the presence, morphology, colour, and type of bacteria. Microscopic evaluation of the Gram-stained smear was used to assess the presence of Gram-positive cocci (*S. suis* is coccoid or ovoid in shape and occurs as single bacteria, as pairs, or in short chains [8]) or other bacteria (Gram-negative rods, potentially expected in pigs with meningitis caused by *E. coli* or *G. parasuis*).

Since normally only one morphotype (bacteria with a certain Gram stain and shape) is seen in a sterile site, only chains of Gram-positive coccoid structures with identical forms and sizes are indicative of the presence of *Streptococcus* spp. [9]. A short chain of *Streptococcus* spp. bacteria was defined as a chain of 5–10 coccoid-shaped bacteria (Figure 2) [9]. As the presence of short chains may not be distributed equally over the smears, every smear was investigated thoroughly in order not to miss possible chains.

Both staining protocols were performed according to the manual provided by the manufacturers. The evaluation was performed by the first author. The impression smears were collected during necropsy and were evaluated retrospectively by the first author after all the samples were collected. Approximately, 25% of the smears (14/54) were randomly selected and were evaluated in a blinded manner by an EBVS-certified veterinary pathologist.

### 2.5. Data Analysis

The microscopic evaluation of the smears was compared with the result of bacterial culture of the leptomeninges, which has been considered the golden standard [10]. Different groups were made based on the abundance of neutrophils in the cytology examination of the smears and based on the presence of short-chain coccoid bacteria. A Fisher’s exact test was used to analyse whether the outcome of the bacterial culture was significantly different between the groups. In case of significant differences between the groups, sensitivity, specificity, and positive and negative predictive values were calculated. Statistical analyses were performed using GraphPad Prism 9 for Windows 64-bit (version 9.0.0 (121), GraphPad Software, San Diego, CA, USA).

## 3. Results

### 3.1. Bacterial Cultures of the Leptomeninges

In 27 of 54 samples of the leptomeninges, a pure bacterial culture was found: *Streptococcus* spp. (*n* = 25), *E. coli* (*n* = 2), and *G. parasuis* (*n* = 1). In nine cases, there was no pure culture: two cases of *E. coli* and *Streptococcus* spp., two cases of mixed culture with the presence of *Streptococcus* spp., and five cases of mixed culture. Bacterial culture was negative in 18 cases.

### 3.2. Presence of Neutrophilic Granulocytes in the Cytologic Impression Smears

The number of smears with either low, moderate, or high abundance of neutrophils assessed with the Hemacolor^®^ Rapid staining was as follows: low (*n* = 19), moderate (*n* = 17), and high (*n* = 18).

The results of the neutrophil abundance in relation to those of the bacterial culture are shown in Table 1. In the group of high abundance of neutrophils (*n* = 18), 17/18 cases were bacteriologically positive, most of the time *Streptococcus* spp. in pure culture (*n* = 10) or *Streptococcus* spp. in combination with other bacteria (*n* = 3). In the group with a moderate abundance of neutrophils (*n* = 17), 14/17 cases were bacteriologically positive, also most of the time *Streptococcus* spp. in pure culture (*n* = 12) or *Streptococcus* spp. in combination with other bacteria (*n* = 1). In the group with a low abundance of neutrophils (*n* = 19), only 5/19 were bacteriologically positive.

### 3.3. Presence of Bacteria in the Cytology Smears

The number of smears with short chains of coccoid bacteria was 12. The presence of short chains in the smears in relation to the results of bacterial culture is shown in Table 2.

### 3.4. Sensitivity, Specificity, and Predictive Values of Microscopic Evaluation of Smears of the Leptomeninges (Abundance of Neutrophils, Presence of Short Chains of Coccoid Bacteria)

The groups based on the abundance of neutrophils were assessed according to the outcome of the bacterial culture. To this end, the groups having moderate and high abundances of neutrophils were combined. Pigs in which one or more bacteria were found in bacterial culture were considered positive (Table 3). Based on Fisher’s exact test, the groups with moderate and high abundances were significantly more likely to have a positive bacterial culture (*p* = < 0.001). This diagnostic test (abundance of neutrophils in the smears) had a sensitivity of 86% (31/36 = 0.861) and a specificity of 78% (14/18 = 0.777) (Table 3). The positive predictive value of smears with moderate or high abundance of neutrophils was 89% (31/35 = 0.885). The negative predictive value of smears with a low abundance of neutrophils was 74% (14/19 = 0.736).

The number of smears with or without short chains of coccoid bacteria was determined according to the outcome of the bacterial culture (Table 4). The result of the bacterial culture was divided based on the presence or absence of *Streptococcus* spp. Based on Fisher’s exact test, the groups with the presence of short chains of coccoid shapes were significantly more likely to be positive with *Streptococcus* spp. in bacterial culture (*p* = < 0.001). The presence of short chains of coccoid bacteria had a sensitivity of 43% (12/28 = 0.428) and a specificity of 100% (26/26 = 1). The positive predictive value of smears with short chains of coccoid bacteria was 100% (12/12 = 1), and the negative predictive value of smears without short chains of coccoid bacteria was 62% (26/42 = 0.619).

## 4. Discussion

The present study demonstrates that microscopical examination of impression smears of the leptomeningeal surface of piglets aged 3 to 10 weeks with neurological signs and/or (acute) mortality is a feasible procedure that can help swine practitioners in quickly establishing a presumptive diagnosis of (per)acute purulent meningitis. A moderate-to-high abundance of neutrophils in the smears had a predictive value for a positive bacterial culture of the leptomeninges of 89%, and a low abundance of neutrophils had a negative predictive value of 74%. Finding moderate-to-high amounts of neutrophils, therefore, points to the bacterial presence within the meninges, which, in principle, can be regarded as proof of bacterial infection since normal liquor and brain tissues are sterile [9,11]. This is expected, as neutrophils are the main immune cell present in bacterial meningitis [11]. In particular, *S. suis* infection is characterised by neutrophilic leukocytosis, and lesions of infected subjects contain a large number of neutrophils, as recently extensively reviewed [12]. Demonstrating the presence of neutrophils may, therefore, assist swine practitioners in the early stages to predict bacterium-induced meningitis and initiate antimicrobial treatment early [13,14]. In addition, the smears were also evaluated for the presence of short chains of coccoid bacteria. In cases in which these were present, they had a 100% predictive value for finding *Streptococcus* spp. in bacterial culture, but when no short chains were found in the smears, the predictive value for the absence of *Streptococcus* spp. in the bacterial culture was only 62%. This means that the presence of short-chain cocci is helpful in cases in which they are seen in the cytological smear and that antimicrobial treatment may even be narrowed down to target *Streptococcus* spp. However, not finding short chains of coccoid bacteria is only a weak predictor for the absence of *Streptococcus* spp. in bacterial culture.

Bacterial culture was chosen as the gold standard in this study. It is a reliable method often performed in normal diagnostic practise to demonstrate the presence of bacteria such as *S. suis* and *E. coli* and thereby used to make the existence of meningitis further plausible. Both *S. suis* and *E. coli* are easy to culture, and they survive post-mortem for a few days [10]. By contrast, the bacterial culture of *G. parasuis* is more difficult, and in addition, the bacterium may die off more quickly in dead piglets. Bacterial culture, however, is only reliable if pigs have not been treated with antimicrobials prior to the examination, which was used as an inclusion criterium in the present study.

Since some diagnostic procedures were performed retrospectively, unfortunately, no brain tissue was available to evaluate the presence of meningitis in addition histologically, because, in regular practise, such samples are often not taken. It seems unlikely, however, that the presence of neutrophils in the context of, for example, generalised leukocytosis would be demonstrable systematically from impression smears of intact, uncut meninges. Furthermore, other known brain lesions to involve marked neutrophilic infiltration, for example, abscesses or localised thrombosis of vessels [15], would likely have been visible macroscopically. However, they were never observed in the selected cases. Taken together with the fact that purulent meningitis is the type of meningitis most often observed in domestic pigs and especially in young individuals [15], it seems very likely that neutrophils that were visualised via the described cytological technique were present freely in the leptomeninges within the context of purulent meningitis.

The necropsy and collection of cytological smears take approximately 15 to 30 min when the technique has become routine, and the proper equipment is present. The staining and evaluation of the cytological smears take an additional 15 to 30 min, depending on experience. This is preferably performed at the veterinary practise. The use of this method on the farm is difficult and unwanted, not simply because of the need for staining materials and a light microscope but also because of the risk of further spreading infectious agents. Opening the skull properly might be challenging. A clean saw to split the skull is needed, and cross-contamination with bacteria from other organs should be avoided. Therefore, it is essential to use a clean place to work, which ideally should be a room specifically appointed to perform necropsies.

Smears with a high abundance of neutrophils could be easily identified, but some experience is needed to evaluate the other groups. There were many smears, especially in the group with a moderate or low number of neutrophils, that contained other more heterogeneous cell clusters. It is known that, especially during an ongoing period of the disease, bacterial infections will lead to infiltration of neutrophils, as well as mononuclear inflammatory cells, e.g., macrophages, lymphocytes, and plasma cells [5,12]. In addition, different *Streptococcus* spp. may induce disease via different pathogeneses and, therefore, a morphologically variable inflammatory response in the animal, which could result in presence of more heterogeneous inflammatory cell populations in the meninges [12,16]. This may also explain the lower predictive values (74%) of smears with a low number of neutrophils for the negative result of bacterial culture, i.e., smears with no or a low number of neutrophils, but where the bacterial culture was positive. Intoxication and viral infections primarily tend to cause non-purulent meningitis, but in later stages of the disease, necrosis or secondary bacterial infections may additionally trigger the influx of neutrophils. It is, therefore, important to consider and ideally further rule out potential underlying non-bacterial disease before blindly starting antimicrobial therapy when finding neutrophils in meningeal smears.

The interpretation of the Gram-stained smears also requires experience, as the short chains of coccoid bacteria are often unevenly distributed in the smears and not easily visible in the cell-dense areas. The whole cytological smear must be evaluated in order not to miss the short chains. The Gram staining may also help to visualise other bacterial agents. It is important to note that bacterial species, whether or not Gram-positive, most of the time cannot be specifically differentiated based purely on their morphology. Therefore, additional techniques, such as bacterial culture, will ultimately always be needed to exactly differentiate between bacterial species and confirm the tentative diagnosis based on smear evaluation. More research is needed to evaluate the usefulness of the Hemacolor^®^ and Gram staining for identifying infections with different bacteria.

Another way of potentially more rapid detection of bacteria than culture could be by performing PCR on cerebrospinal fluid (CSF) [11,17,18]. However, veterinary practises usually do not have such equipment available. In addition, this would also require sending samples to external laboratories, and therefore, the time advantage would be lost. On the other hand, for demonstration of bacteria that are difficult to culture, for example, *G. parasuis*, PCR could be a useful method to confirm the diagnosis.

Apart from bacterial or other infections, neurological signs may also result from non-infectious factors such as water deprivation (salt intoxication) or toxins damaging the endothelium of blood vessels (e.g., oedema disease). In these cases, opisthotonos is often absent or much less pronounced, and neutrophil infiltration is often very minimal or less abundant. Other diagnostic procedures or criteria are needed to establish the diagnosis of these diseases [19]. The major part of the mortality and morbidity in nursery pigs, however, is related to bacterial infections. At the Royal Animal Health Service in the Netherlands, more the 15% of the necropsy cases in pigs are related to *S. suis* [2]. Similar data are known from the United States [20]. This prevalence of *S. suis* infection is reflected in the present study, as 67% of the samples from pigs with neurological signs and/or (acute) mortality were positive for *Streptococcus* spp. on bacterial culture.

## 5. Conclusions

The abundance of neutrophilic granulocytes in an impression smear of the leptomeningeal surface of piglets with neurological signs and/or (acute) mortality was highly predictive of a positive bacterial culture of the leptomeninges. In addition, finding short chains of coccoid bacteria in Gram-stained smears was 100% predictive of a positive culture of *Streptococcus* spp. This study demonstrates that cytological examination of smears of the meninges is a feasible procedure that can help swine practitioners in establishing a tentative diagnosis of bacterial meningitis, in order to quickly initiate better evidence-based treatment to avoid further animal suffering and mortality when awaiting the results of bacterial culture.

## Figures and Tables

**Figure 1 vetsci-09-00341-f001:**
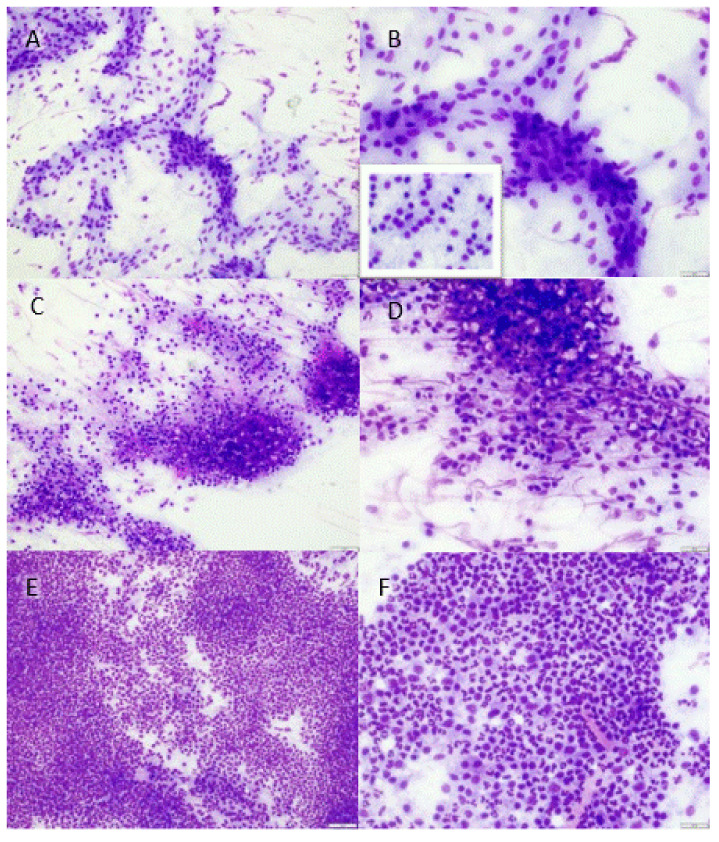
Hemacolor^®^ staining of leptomeningeal smears of pigs, shown according to abundance of neutrophils: (**A**,**B**) low, (**C**,**D**) moderate, and high (**E**,**F**). (**A**,**B**) Monomorphic flat to sometimes cuboidal (inset) cells within small amounts of extracellular matrix (fibroblasts with collagen). Neutrophils are rarely present (<5 per 400× magnification area) and are not visible in the selected areas for the photo panel (200× (**A**) and 400× (**B**)); (**C**,**D**) areas with neutrophils (5–100 per 400× magnification area), mixed with areas of fibroblasts and collagen and sometimes other, more heterogeneous cell clusters (likely mononuclear inflammatory cells) (200× (**C**) and 400× (**D**)); (**E**,**F**) large numbers of neutrophils (>100 to ‘uncountable’), obscuring other potentially present cells (200× (**E**) and 400× (**F**)).

**Figure 2 vetsci-09-00341-f002:**
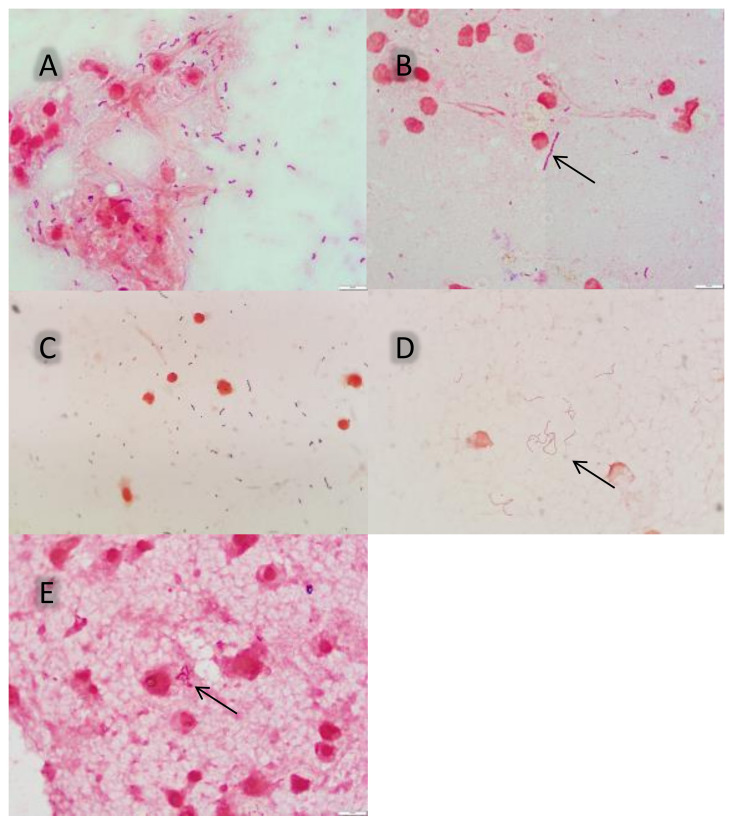
Gram staining of smears of the leptomeninges of pigs: (**A**) multiple Gram-positive coccoid shapes with short chains (5–10 coccoid shapes). Bacterial culture of the leptomeninges was positive for *Streptococcus* spp. (1000×); (**B**) a chain (arrow) with 18 coccoid shapes (1000×). Bacterial culture of the leptomeninges was positive for *Streptococcus* spp; (**C**) multiple short chains with Gram-positive coccoid shapes. Bacterial culture of the leptomeninges was positive for *Streptococcus* spp. (1000×); (**D**) multiple long chains (>10 coccoid shapes) (arrow). Bacterial culture of the leptomeninges was positive for *Streptococcus* spp. (1000×); (**E**) rod-shaped, Gram-negative bacteria (arrow). Bacterial culture of the leptomeninges was positive for *G. parasuis* (1000×).

**Table 1 vetsci-09-00341-t001:** Abundance of neutrophils in cytologic impression smears from leptomeninges of pigs (*n* = 54) in relation to the results of bacterial culture.

		Abundance of Neutrophils in Smear			
Bacterial Culture		High (*n* = 18)	Moderate (*n* = 17)	Low (*n* = 19)	Total
Negative		1	3	14	18
Positive	*Streptococcus* spp.	10	12	2	24
	*E. coli*	1	0	1	2
	*G. parasuis*	1	0	0	1
	*E. coli* and *Streptococcus* spp.	2	0	0	2
	Mixed culture and *Streptococcus* spp.	1	1	0	2
	Mixed culture	2	1	2	5

**Table 2 vetsci-09-00341-t002:** Presence of short chains of coccoid-shaped bacteria in cytologic impression smears of the leptomeninges of pigs in relation to the results of bacterial culture of the leptomeninges.

		Short Chains of Coccoid Shaped Bacteria in Smear		
Bacterial Culture		Present (*n* = 12)	Absent (*n* = 42)	Total
Negative		0	18	18
Positive	*Streptococcus* spp.	10	14	24
	*E. coli*	0	2	2
	*G. parasuis*	0	1	1
	*E. coli* and *Streptococcus* spp.	0	2	2
	Mixed culture and *Streptococcus* spp.	2	0	2
	Mixed culture	0	5	5

**Table 3 vetsci-09-00341-t003:** Abundance of neutrophils (high–moderate versus low) in a cytological smear in relation to the bacterial culture of the leptomeninges (either positive or negative) *.

	Bacterial Culture of Leptomeninges		
Abundance of Neutrophils in Smear of the Leptomeninges	Positive	Negative	Total
High and moderate	31	4	35
Low	5	14	19
**Total**	36	18	54

* Two-sided Fisher’s exact test: *p*-value was < 0.001.

**Table 4 vetsci-09-00341-t004:** Presence of short chains of coccoid bacteria (yes versus no) in a cytological smear of the leptomeninges in relation to the bacterial culture of the leptomeninges (*Streptococcus* spp. present or not) *.

	*Streptococcus* spp. Found in Bacterial Culture		
Presence of Short Chains of Coccoid Shapes	Yes	No	Total
Yes	12	0	12
No	16	26	42
**Total**	28	26	54

* Two-sided Fisher’s exact test: *p*-value was < 0.001.

## Data Availability

The dataset supporting the conclusions of this article is included within the article as Appendix A. A table is added with all data collected from each pig: a brief description of macroscopical findings, the microscopical evaluations, and the results of bacterial culture.

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
