# Peer review of "Microscopical Evaluation of Smears of the Leptomeninges to Predict Meningitis in Piglets"

_vetsci, 2022, doi:10.3390/vetsci9070341_

Round 1
Reviewer 1 Report
The manuscript entitled “Microscopical evaluation of smears of the leptomeninges to predict meningitis in piglets” is well structured and well written by the authors.
The "introduction" section is consistent with the experimental work, accompanied by relevant and current bibliography and sufficiently extensive.
The "materials and methods" section is properly divided into subsections each with a detailed to clear description of the procedures.
The results are well argued and reported with clarity and completeness.
The discussion is very explanatory although comparisons with other similar work that has been conducted on the same topic in recent years could be added and would further enrich the discussion by greatly enhancing it.
I cannot imagine what the opinion of other reviewers is, but, in my opinion, the table reported by the authors as supplementary material could absolutely be included and commented on within the main body of the manuscript.
Finally, I advise Authors to follow these few directions and make the requested additions and suggest, after complying with what is required, to proceed with publication.
Author Response
Response to Reviewer 1 Comments
The "introduction" section is consistent with the experimental work, accompanied by relevant and current bibliography and sufficiently extensive.
The "materials and methods" section is properly divided into subsections each with a detailed to clear description of the procedures.
The results are well argued and reported with clarity and completeness.
The discussion is very explanatory although comparisons with other similar work that has been conducted on the same topic in recent years could be added and would further enrich the discussion by greatly enhancing it.
We have rewritten the discussion and added relevant recent work, see the new text of the article that is uploaded.
I cannot imagine what the opinion of other reviewers is, but, in my opinion, the table reported by the authors as supplementary material could absolutely be included and commented on within the main body of the manuscript.
We can understand that adding this table to the main text could have some added value. However, we would prefer not to do so. One of the main reasons is that all this large amount of information would dilute the main message and greatly reduce the readability of the article. The main message can be better presented without the need to add the extra (interesting, but not key) information. The possible few readers that are interested in this information, can read the supplementary material. Also this request was not mentioned in the evaluations of the other reviewer and the editor.
Finally, I advise Authors to follow these few directions and make the requested additions and suggest, after complying with what is required, to proceed with publication.

Reviewer 2 Report
The short paper from Schyns et al describes the microscopical evaluation of meningitis in piglets at necropsy using cytological smearing and staining and bacterial culture. The paper is well-written and sound presentation of data that there is little of concern. Some minor questions for discussion -
1. How long does it take to do the whole procedure form accepting the piglet for necropsy through to confirmation that it is meningitis?
2. Why not take a CSF sample by lumbar puncture and do PCR for bacteria? Can you discuss the pros and cons of doing this as a protocol. It might be easier and quicker to take CSF than split a skull and sample the meninges. Or is there a practical issue in taking CSF from dead piglets?
3. What sort of diseases (non-bacterial) would result in neutrophils in the meninges? Line 302-303, clarify.
4. references - no.2? No.5 page numbers.
Author Response
Response to Reviewer 2 Comments
The short paper from Schyns et al describes the microscopical evaluation of meningitis in piglets at necropsy using cytological smearing and staining and bacterial culture. The paper is well-written and sound presentation of data that there is little of concern. Some minor questions for discussion -
- How long does it take to do the whole procedure form accepting the piglet for necropsy through to confirmation that it is meningitis? `
The necropsy and collection of cytological smears takes approximately 15 to 30 minutes when the technique has become routine and the proper equipment is present. The staining and evaluation of the cytological smears takes an additional 15 to 30 minutes depending on experience (see lines 312-315). - Why not take a CSF sample by lumbar puncture and do PCR for bacteria? Can you discuss the pros and cons of doing this as a protocol. It might be easier and quicker to take CSF than split a skull and sample the meninges. Or is there a practical issue in taking CSF from dead piglets?
Another way to detect bacteria potentially quicker than via culture could be by performing PCR on cerebrospinal fluid (CSF) [11,17,18]. Veterinary practices usually do not have such equipment available, however, and this would require sending samples to external laboratories and therefore the time advantage would be lost. On the other hand, for demonstration of bacteria that are difficult to culture, for example G. parasuis, PCR could be a useful method to confirm the diagnosis (see lines 349-355). - What sort of diseases (non-bacterial) would result in neutrophils in the meninges? Line 302-303, clarify.
1. Intoxication and viral infections primarily don’t cause a large influx of neutrophils in the meninges, they primarily tend to cause non-purulent meningitis. However, in later stages of disease necrosis or secondary bacterial infections might additionally occur before acute death. Immune responses and ubiquitous bacteria that become septic can be responsible for influx of neutrophils. It is therefore important to consider and ideally further rule out potential underlying non-bacterial diseases before blindly starting antimicrobial therapy when finding neutrophils in meningeal smears (see lines 333-348).
2. Regarding this question, we now also have discussed the fact that no additional histological examination of tissues could be performed, due to the fact that this was some diagnostics were done retrospectively on field cases. When these cases originally were evaluated diagnostically, no appropriate samples for such examination were taken, because this is normally often not done in practise with these kind of cases (only when officially sent to external laboratories for necropsy by certified pathologists). Main section discussing this aspect is added in lines (300-311) and small textual changes are made throughout the Introduction and Material and Methods sections to address this more properly. - references - no.2? No.5 page numbers.
2. Geudeke, T.; Duinhof, T.; Heuvelink, A. Monitoring Diergezondheid Varkens - Rapportage Tweede Halfjaar 2019; 2019;
5. Dunbar, S.A.; Eason, R.A.; Musher, D.M.; Clarridge Iii, J.E. Microscopic Examination and Broth Culture of Cerebrospinal Fluid in Diagnosis of Meningitis. J. Clin. Microbiol. 1998, 36, 1617–1620.
